# Thermography of Asteroid and Future Applications in Space Missions

**Tatsuaki Okada**

Institute of Space and Astronautical Science, Japan Aerospace Exploration Agency, 3-1-1 Yoshinodai, Chuo, Sagamihara 252-5210, Japan; okada@planeta.sci.isas.jaxa.jp; Tel.: +81-50-3362-2471

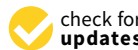

**Featured Application: Activities in lunar, planetary, and asteroid science and exploration.**

**Abstract:** The Near-Earth Asteroid 162173 Ryugu is a C-type asteroid which preserves information about the ancient Solar System and is considered enriched in volatiles such as water and organics associated with the building blocks of life, and it is a potentially hazardous object that might impact Earth. Hayabusa2 is the asteroid explorer organized by the Japan Aerospace Exploration Agency to rendezvous with the asteroid and collect surface materials to return them to Earth. Thermography has been carried out from Hayabusa2 during the asteroid proximity phase, to unveil the thermophysical properties of the primitive Solar System small body, which offered a new insight for understanding the origin and evolution of the Solar System, and demonstrated the technology for future applications in space missions. Global, local, and close-up thermal images taken from various distances from the asteroid strongly contributed to the mission success, including suitable landing site selection, safe assessment during descents into the thermal environments and hazardous boulder abundance, and the detection of deployable devices against the sunlit asteroid surface. Potential applications of thermography in future planetary missions are introduced.

**Keywords:** thermography; uncooled micro-bolometer array; asteroid; planetary exploration; thermal inertia

## 1. Introduction

The Thermal Infrared Imager TIR [1] (see Figure 1, Table 1) is a remote sensing instrument on Hayabusa2, the Japanese second sample return mission from the asteroid [2], which is based on a two-dimensional uncooled micro-bolometer array with $328 \times 248$ effective pixels, with $16.7° \times 12.7°$ field of view, and covering 8–12 μm wavelength range using a germanium lens and a band pass filter [1]. The absence of a cooling system in the TIR contributes to its small, lightweight, and long-living design. The basic design of the sensor unit (TIR-S), the digital electronics (DE), and the onboard data processing algorithm is inherited from the Longwave Infrared Camera from the Akatsuki Venus mission [3], but the wide detection range of temperature, observation sequences and plans, programming of data processing, and calibration methods are optimized for the TIR [4]. As with a deep-space mission, the maximum downlink rate is limited to within 32 Kbps, and onboard processing is essential for its design [1,2].

Asteroids are primitive bodies of the Solar System and probably the parent bodies of meteorites. Planetary formation started with fluffy dust in the solar nebula, followed by accumulation of dust to planetesimals, or re-accretion to rubble-piles from fragments by meteoroid impacts. The formation and evolution processes will be indicated by the surface physical state of asteroids or planets, such as grain size, porosity, boulder abundance, and roughness, which are informed by thermal infrared observations. Thermophysical models of asteroids [5] have been investigated and constructed for

interpreting those ground observations in thermal infrared wavelengths, but more detailed models are needed for quantitative analysis using data from a spacecraft [1,4].

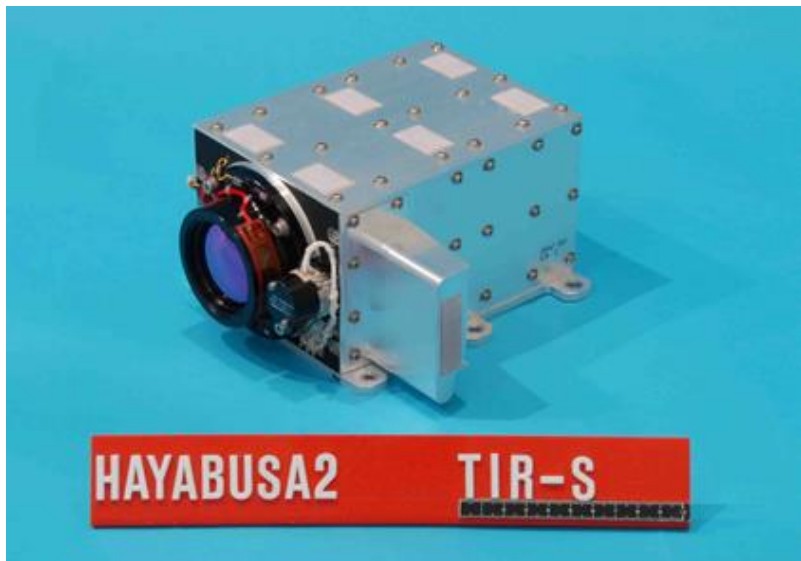

**Figure 1.** The flight model of the sensor unit of the thermal infrared imager TIR onboard Hayabusa2 (TIR-S). White squares on the sensor case are the Velcro tapes to attach the multi-layer insulators (MLI, the thermal blankets, not shown) which cover the sensor unit to stabilize the temperature during the operation. A baffled sunshade (not shown) is attached in front of the germanium lens to avoid insolation. The aperture of the lens is 42 mm in diameter (diameter of lens is 47 mm), and the stepping motor for the mechanical shutter is seen at the right hand of the lens. The sheet heater around the lens case is to adjust the temperature of the optics. The scale on the name plate is 10 cm long (1 cm each). (©Japan Aerospace Exploration Agency (JAXA)).

**Table 1.** Characteristic performance of TIR on Hayabusa2 [1].

| Items | Performance |
|---|---|
| Total Mass | 3.28 kg (DE is not included) |
| Total Power | 18 W (nominal) (DE is not included) |
| Detector | Uncooled bolometer array NEC 320A (VO) (anti-reflection coating) |
| Pixels | 344 × 260 (effective 328 × 248) |
| Field of View (FOV) | 16.7° × 12.7° |
| IFOV | 0.89 mrad (0.051°) |
| Aperture | 42 mm (effective), lens diameter is 47 mm |
| MTF (@Nyquist Freq.) | 0.5 |
| F-number | 1.4 |
| Temperature range | 233–423 K (well calibrated), 150–460 K (detectable range) |
| NETD | <0.3 |
| Absolute Temperature Range | <3 K |
| A/D Converter | 12 bit (15 bit after summed) |
| Reference Temperature | Shutter temperature (monitored, nominally at 301 K) |
| Frame Rate | 60 Hz |
| Image Summation | 2ˆm, m = 0 to 7 (1,2, ... ,128) |
| Consecutive shot numbers | Programable (max 128 images), current programed: 30 |

Remote thermal radiometry is an often-used technique in planetary missions [6–11] to investigate the physical state of a planetary surface layer, especially in its particle size distribution of regolith and boulder abundance, which are derived from thermal inertia, $\sqrt{\rho k c_p}$, with density $\rho$, thermal conductivity $k$, and heat capacity $c_p$. Thermal inertia is an index of thermophysical properties that

tend to have lower thermal inertia for more porous material, while higher thermal inertia for the denser material. Typical examples show that for a low thermal inertia below 50 J m$^{-2}$ s$^{-0.5}$ K$^{-1}$ (tiu, hereafter), the surface is composed of fine-grained materials such as lunar regolith, while for a high thermal inertia beyond 2000 tiu, the surface consists of non-porous dense rocks [1]. On the other hand, carbonaceous chondrite meteorites that are primitive and enriched in volatiles are likely analogues to C-type asteroids [12,13], which show an intermediate thermal inertia between 600 to 1000 tiu [14].

Thermal inertia can be derived from thermal measurements, since diurnal temperature profiles show a large peak-to-peak temperature difference for the case of lower thermal inertia, while a temperature change becomes much smaller and the peak temperature is delayed from the local noon for the case of higher thermal inertia. For the Moon, Mars, large asteroids, and satellites, the surface is covered with fine-grained or graveled regolith that is ejected by hypervelocity meteoritic impacts and sedimented on the surface. However, an in-depth study is required for small bodies since their surface condition has been poorly understood [15] since most of the ejecta could be escaped and the basement might be exposed due to low gravity.

Hayabusa2 is the first mission to investigate the surface physical state of a small body through thermography [4]. In past planetary missions, thermal radiometry has been mainly conducted along the track of the orbiting spacecraft from a low circular orbit. However, thermal imaging is requested for Hayabusa2 because the spacecraft was not orbiting the asteroid but staying at a distant stational position, so that the whole surface was imaged by asteroid rotation to determine the temperature distribution and its diurnal temperature profiles of each geologic site. In Section 2, the Hayabusa2 and the asteroid Ryugu are introduced in more detail. In Section 3, the Hayabusa2 results during the approaching phase are displayed. In Section 4, the Hayabusa2 results during the proximity phase are shown, and in Section 5, the technical applications in the future missions is introduced and discussed, and in Section 6, the concluding remarks of this paper are described.

## 2. Hayabusa2 and Asteroid 162173 Ryugu

Hayabusa2 [2] is the second asteroid mission organized by Japan Aerospace Exploration Agency (JAXA), inherited from the Hayabusa mission [16] which visited and returned samples from S-type asteroid 25143 Itokawa. Hayabusa2 is to explore a C-type asteroid, which is considered to be a parent body of volatile-rich carbonaceous chondrite meteorites and a key target to investigate the origin and evolution of the early solar system and the possible source materials of building blocks of life [17].

Ground-based and space-based observations [18] predicted that Ryugu is classified as a C-type asteroid in taxonomy, of rounded shape with a diameter of about 0.9 km, and with rotation in 7.63 h. It is basically dark material with geometric albedo below 0.045, and possibly covered with coarse regolith of centimeter-sized granules with average thermal inertia of 150 to 300 tiu. Before the arrival of the Hayabusa2 spacecraft at Ryugu, the asteroid was considered to be a desiccated carbonaceous chondrite meteorite after thermally and aqueously alteration, and the surface might be sedimented with coarse-grained regolith materials and dense boulders, as well as excavated by impact craters where the materials from the interior might be excavated and exposed.

Hayabusa2 was launched on 3 December 2014 and arrived at the home position, 20 km earthward from Ryugu on 27 June 2018, and started remote sensing [19] using the optical navigation camera (ONC) [20], the laser altimeter (LIDAR) [21], the near infrared spectrometer (NIRS3) [22], and the TIR [1,4] to characterize the asteroid regarding its shape, spin state, geomorphology, spectral and thermal properties, geologic features, and gravity. The state and characteristics of Ryugu before and after arrival was basically consistent, but the asteroid shape is double top-shaped, and the surface is not covered with regolith but with boulders, and no flat and smooth terrains larger than 10 m scale were found for easy and safe landing [4,19–22].

### 3. TIR Observations of Ryugu in Cruise and Approach Phases

During a 3.5-years long cruise phase before arrival at Ryugu, images of deep sky backgrounds were taken several times per year as the dark frames for calibration, and the degradation of TIR was checked by comparing the temporal changes of its bias levels and the standard deviations as well as the number of bad pixels [4,23]. No substantial degradation has been found for five years after launch, so that the radiation tolerance of the TIR against the deep-space environment has been proven.

Hayabusa2 passed by Earth for the gravity assist to change its trajectory en route to Ryugu on 3 December 2015. From 14 October 2015, the spacecraft changed its attitude to point Earth and the Moon to observe them with the remote sensing instruments until 21 December 2015. During this period, TIR has taken images of Earth and the Moon from the various distances, so that the dependency of distance on thermal radiation intensities has been investigated [23]. The critical distance above the threshold level for detecting Earth and the Moon were obtained, which was applied to the estimate of the critical distance ($-1.5 \times 10^4$ km) or the critical size of diameter ($-0.06$ mrad) at 1 au from the Sun for detecting Ryugu by TIR before arrival [23].

The first light event of TIR for the observation of Ryugu was conducted on 6 June 2018 at $2.5 \times 10^3$ km from the asteroid. Ryugu was successfully detected as was predicted [23]. A series of one-shot imaging of Ryugu by TIR continued almost once a day until arrival at the asteroid (see Figure 2, Table 2) and a relationship of thermal radiation intensity to distance from the asteroid was obtained, as well as to distance from the Sun and the solar phase angle (nearly Sun–Probe–Earth angle). It was important that Ryugu was the only bright spot detected in the thermal images after subtraction of dark frames, which is different from optical images where numerous bright spots other than the target body were found such as bright background stars, bad pixels, or by irradiation. In addition, thermal images by TIR are never saturated by overexposure for most planetary surfaces (< 460 K). These features have an advantage of using thermal images for the survey of orbiting satellites around Ryugu or possible ejection of materials from Ryugu if they were larger than the critical size of diameter [23]. Such satellites or ejection have not been detected through the asteroid proximity phase, although some ejected flows were observed on the asteroid Bennu by the OSIRIS-Rex mission [24].

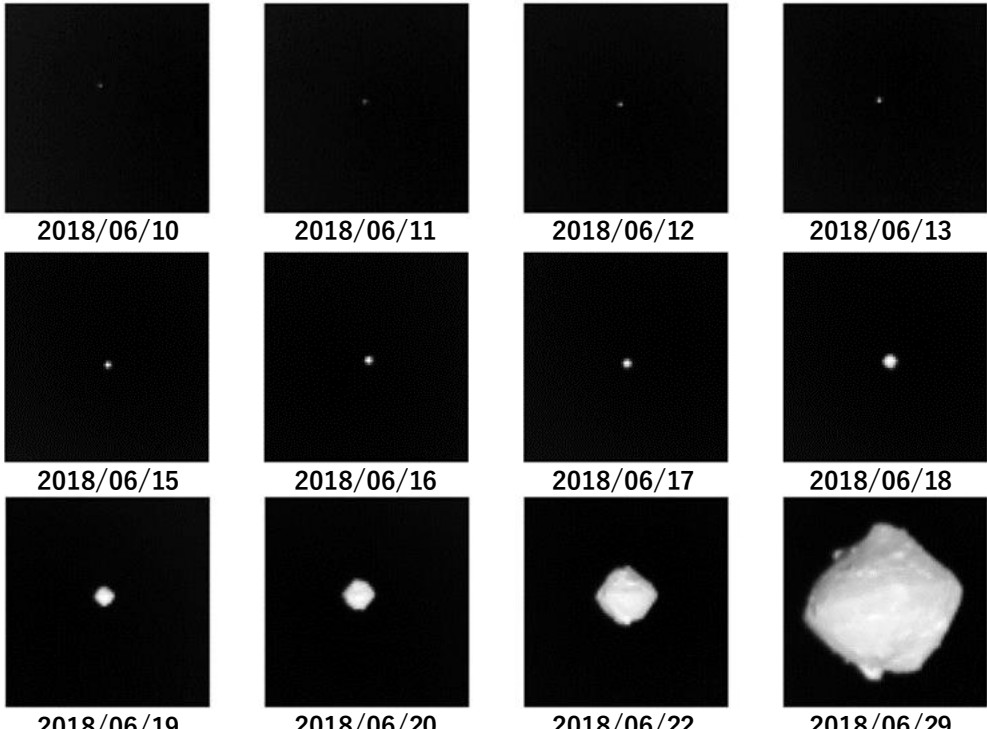

**Figure 2.** Thermal images of asteroid 162173 Ryugu taken by TIR during the Approach Phase (©JAXA).

**Table 2.** Daily observations of asteroid Ryugu by TIR during the Approach Phase.

| Date of Observation | Distance from Sun [au] | Sun–Probe–Earth Angle [deg] | Distance from Ryugu [km] |
|---|---|---|---|
| 10 June 2018 | 0.9650 | 17.02 | 1550 |
| 11 June 2018 | 0.9655 | 17.09 | 1380 |
| 12 June 2018 | 0.9661 | 17.20 | 1150 |
| 13 June 2018 | 0.9667 | 17.28 | 1000 |
| 15 June 2018 | 0.9681 | 17.45 | 600 |
| 16 June 2018 | 0.9689 | 17.53 | 500 |
| 17 June 2018 | 0.9698 | 17.63 | 350 |
| 18 June 2018 | 0.9707 | 17.71 | 200 |
| 19 June 2018 | 0.9716 | 17.78 | 170 |
| 20 June 2018 | 0.9727 | 17.87 | 100 |
| 22 June 2018 | 0.9749 | 18.01 | 49 |
| 29 June 2018 | 0.9843 | 18.48 | 20 |

## 4. TIR Observations in Asteroid Proximity Phase

The Asteroid Proximity phase started on 28 June 2018 at the home position for almost 1.5 years, and Hayabusa2 kept its position there during the phase. The operations were categorized into three types: (1) Keeping the position within safety boxed areas, for the global and oblique angle observations; (2) Controlled descent operations down to −1 km altitude, for high-resolution local thermal imaging; and (3) Descent operations for sampling or lander release, for close-up thermal imaging.

### 4.1. Global Thermal Images of Ryugu

The first global disc-resolved thermal image set of an asteroid in history was taken by TIR for one asteroid rotation on 30 June 2018, just after arrival at the asteroid. The surface temperature in the daytime was around 300 to 370 K at a solar distance of 0.986 au, at the Sun–Probe–Earth (SPE) angle of 18.6°. Comparison with calculated thermal images [25,26] shows that the surface of Ryugu is colder than in the case of fine regolith surface where the thermal inertia is about 50 tiu and hotter than in the case of base rocks whose typical thermal inertia is higher than 1000 tiu [4,20]. It was interesting that the temperatures of large boulders with several tens of meters or larger size on Ryugu are almost equal to those of the surrounding surfaces [4]. This was inconsistent with the original idea before arrival, where the large boulders should have high thermal inertia larger than 1000 tiu and looks like "cold spots", i.e., much colder than the surrounding [4].

It was also found that the diurnal temperature profiles of the surface of Ryugu are rather constant compared with the anticipated profiles for a simple thermal calculation where a one-layer homogeneous material and a flat surface without roughness being assumed [4]. The thermal inertia of most geologic sites on Ryugu ranged around 300 ± 100 tiu [4,26], which was derived from the maximum temperature and its delay from the local noon. This implies that the surface physical state of Ryugu is consistent with very porous and rough surface, where boulders have high porosity (30% to 50 %) and the rest of the surrounding surface is dominated by fragments of porous rocks several centimeters larger than the thermal skin depth [4]. It is important that the surface physical state predicted from the global thermal images was confirmed later by the close-up thermal imaging by TIR and also by the optical surface imaging and thermal radiation measurements from the surface lander Mobile Asteroid Surface Scout (MASCOT) [27,28].

Most thermal images are taken from the low SPE angles but there are large solar phase angle observation campaigns to view the asteroid from the oblique direction. Figure 3 shows an example of the oblique thermal image of Ryugu taken by TIR at SPE angle of 42°, at rotation phase of 105°E, where the largest crater Urashima was found at equator of the asteroid (center-left). The flat diurnal temperature profiles and the temperature gradient at the subsolar to the night side should be well investigated since they are influenced by the roughness effect [26].

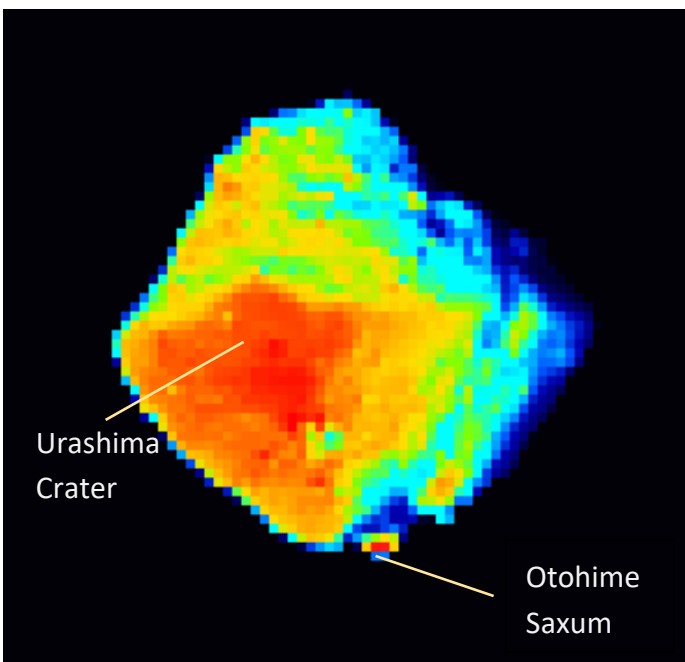

**Figure 3.** Pseudo-color oblique thermal image of Ryugu on 31 August 2018 taken by TIR from the dusk side (SPE angle of 42°). The largest crater Urashima (diameter of 270 m) is seen at the equator, and the largest boulder Otohime is seen at the south pole. The night region shown as the blue color area in the right end of the asteroid is also seen in the thermal image. (Image: hyb2_tir_20180831_130142_l1.fit, ©JAXA).

## 4.2. High Resolved Global and Local Thermal Images of Ryugu

Higher-resolution global and local thermal images of Ryugu were taken by TIR from 5 km altitude several times during the asteroid proximity phase [4,26], and on the way to the lower altitude down to 1 km altitude for the gravity measurement campaign on 6 to 7 August 2018, and during the hovering operation at 3 km altitude after the release of MASCOT lander on 3 to 4 October 2018 [4].

A high-resolution global thermal image set of Ryugu was taken on 1 August 2018 during the Mid-Altitude Observation Campaign, from the altitude of 5 km, at the solar distance of 1.056 au, and from the SPE angle of 19.0°. Figure 4 shows a thermal image at rotation phase of 120°E. Its spatial resolution is 4.5 m per pixel. In this image, surface geologic features such as large boulders are distinguished from the surroundings more clearly than in the images at the home position. Large boulders of tens of meters scale are clearly identified one by one in the thermal images and their temperatures are almost the same as the surroundings. Despite temperature variation for each of the boulders, craters, or other geologic features, no "hot spots" nor "cold spots" are apparently identified in the thermal images of this spatial resolution.

Thermal images show almost homogeneous temperature distribution on a boulder whose surface has both an apparently sedimented area and an apparently non-sedimented bare rock surface area, indicating that the sediments have the same thermal inertia with the surroundings. This implies that the sediments should not be fine regolith made of soils and granules but probably fragments of porous boulders with the same thermal inertia as the huge boulders, whose size must be larger than the thermal skin depth. [20,26].

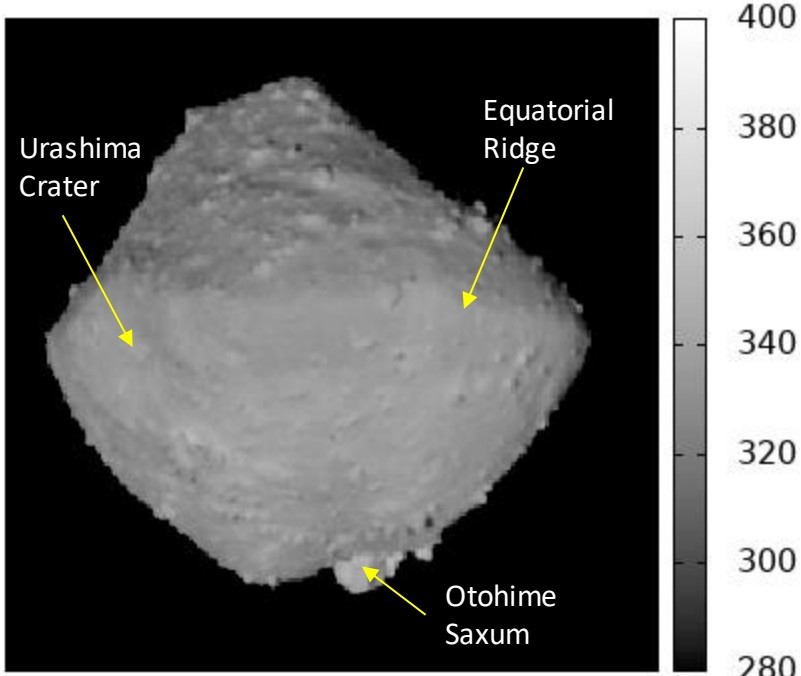

**Figure 4.** Thermal image of Ryugu taken by TIR on 1 August 2018 from the Mid-Altitude (5 km from Ryugu), with the spatial resolution of 4.5 m per pixel. The largest crater Urashima is observed at the equator, and the largest boulder Otohime is seen at the South Pole. Boulders larger than 20 m are all identified in this image set, showing almost the same temperature with the surroundings. The temperature scale is in Kelvin. (Image: hyb2_tir_20180801_152032_l2.fit, ©JAXA).

### 4.3. Close-up Thermal Images of Ryugu

Close-up thermal images of local sites of Ryugu have been taken by the TIR at the altitudes below 100 m during the release of landers and during sampling activities. Those close-up thermal images are the first data sets in planetary missions. Close-up thermal images show the surface physical state and a variety of boulders at centimeter scale. As was seen by optical navigation cameras and the surface landers, it was evident that the surface of Ryugu is widely covered with several centimeter- to meter-sized rocks and boulders, but not with fine regolith [20,27,28]. The surroundings in the global and local thermal images correspond to the rough surface terrains that are covered with boulders down to the tens-of-centimeter scale in the close-up thermal images (see Figure 5).

These close-up thermal images have proven that the thermal inertia of large boulders and their surroundings are almost the same, since the surroundings consist of rocks larger than the typical thermal skin depth of centimeter scale. We also recognized several boulders as "cold spots", remarkably colder by more than 20 K compared to the surroundings [4,26]. The "cold" boulders should be the dense rocks with lower porosity, and probably have typical thermal inertia of carbonaceous chondrite meteorites of 700 to 1000 tiu [14].

As shown in Figure 5, close-up thermal images also show that surface boulders have fine structure in them, indicating rugged rocks with very low thermal inertia of < 300 tiu. Their porosity should be more than 30 % (probably −50%) [4]. These low-density and high-porosity boulders indicate different origin from terrestrial rocks or even meteorites. Considering the planetary formation processes, the formation started with fluffy dust in the solar nebula, and bodies grew by coagulation and accumulation to form planetesimals. Continued accumulation resulted in larger bodies to become the parent body of Ryugu. It should be a porous body because it has never experienced thermal alteration nor igneous processes, except for a low-level consolidation. At the innermost layer of the body, materials should be more consolidated due to higher lithostatic stress and temperatures. After impact fragmentation, the fragments are accreted together to form the present-day asteroid Ryugu. That is why most of

surface materials are highly porous, except for dense rocks from higher consolidated materials such as meteorites. They should be sediments of impact fragments slightly consolidated by the lithostatic stress in the deep inside of the parent body. It cannot be ruled out that the dense boulders might have originated from the impacted meteoroid and have survived during the impact fragmentation processes [4].

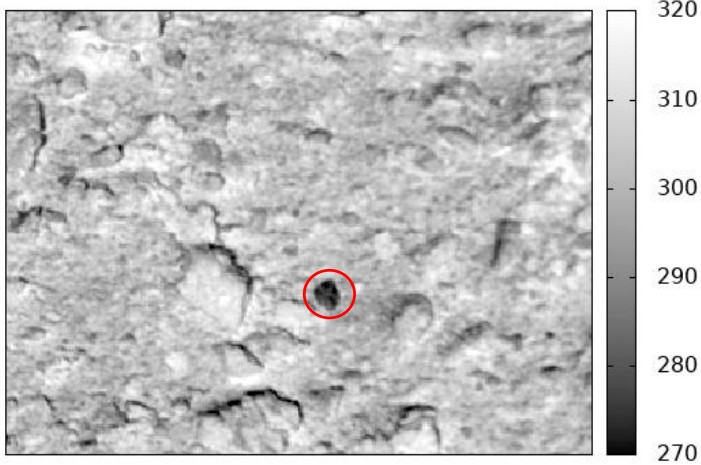

**Figure 5.** Close-up thermal image taken by TIR on 25 October 2018, during the descent operation of TD1-R3 Campaign. The width of this image is 20 m across. Most of boulders are the same temperature but a cold boulder (in a red circle) is seen as a "Cold Spot". The temperature scale is in Kelvin. (Image: hyb2_tir_20181025_022416_l2.fit, ©JAXA).

## 5. Applications of Thermal Images to Future Space Missions

It has been proven that thermal infrared imaging does not only give a strong contribution to understanding the physical state of planetary surfaces but also shows potential applications to future space missions such as optical navigation especially in the nighttime ore dawn/dusk regions, as well as a hazard detection. Here we show some examples of the applications.

### 5.1. Optical Navigation for Approaching the Asteroid

Thermal imaging is applicable to the optical navigation for approaching the target body during rendezvous or fly-by operations. After extracting dark frame images, the target asteroid Ryugu was a one pixel-sized body that was taken during the first light event of Ryugu on 6 June 2018 at a distance of $2.5 \times 10^3$ km, and it became larger and brighter until the arrival at the home position. It was predicted that the asteroid was able to be detected by the TIR at a distance within $1.5 \times 10^4$ km, comparing the relationship of thermal radiation intensity to distance for Earth–Moon observations in 2015 [23]. Thermal images have an advantage in optical navigation compared with optical images by ONC, since there are no other bright spots in the images such as background bright stars, bad pixels, or irradiation of solar wind particles or galactic rays, and there is no need to change the exposure time to avoid overexposure when the target is seen from a single pixel until a numerous pixel-sized body. Onboard functions of dark frame subtraction and binarization implemented in TIR allow optical navigation by data delivery to the guidance and navigation control system of the spacecraft.

### 5.2. Detection of Meteoroids and Moons

As was mentioned in 5.1, the fact that an object is seen as the only bright spot in thermal image enables detection of the meteoroids or the orbiting moons which encounter nearby the spacecraft. Attempts to detect the meteoroids around the Sun–Earth Lagrange points were conducted in April 2016, using the optical imagers the ONC and TIR. The ONC has higher spatial resolution by one order of magnitude and the lower detection limit of the size of meteoroids by about one order of magnitude,

but the ONC images have numerous bright spots other than the target. The TIR can detect meteoroids confidently if they exist within the distance for its detectable size; within $1.5 \times 10^4$ km for 1 km-sized body, $1.5 \times 10^2$ km for 10 m-sized body, or 1.5 km for 10-cm sized body, provided that the meteoroids have similar geometric albedo and emissivity to Ryugu [23]. For the case of S-type asteroids which have several times higher geometric albedo, the detection range is a little bit smaller, but there is no substantial difference in the detection range. A similar situation has occurred during the approach and proximity phases for the survey of orbiting moons around the asteroid for the both purposes of asteroid science and spacecraft safety. It is difficult to detect natural moons with optical images because of saturation by the presence of the very bright main large asteroid and its smearing and blooming effect. Thermal images have an advantage of searching moons, since there is no need to change the exposure time to avoid saturation, smearing, or blooming effect. Unfortunately, TIR has never found any meteoroids at the Sun–Earth Lagrange points nor orbiting moons around Ryugu.

### 5.3. Target Maker Tracking

Close-up thermal imaging during descent operations of Hayabusa2 was carried out using the TIR. An opportunity arose to release and track a Target Marker (TM) on 25 October 2018 (see Figure 6). A TM is a 10 cm-diameter spherical ball wrapped with a recursive reflection sheet, and it has been proven that the TIR clearly detects the TM falling to and settling on the asteroid surface as a "cold sphere". The TM was not clearly detected by optical images at the solar phase angle of > 2° without a flashlight, and if the TM was seen, it tended to be dazzling when a flashlight was used. The first sampling of the asteroid surface material was carried out on 22 February 2019, where the TM was detected again by the TIR as a "cold sphere" beside a 1 m-sized boulder nearby the landing site. The known size and shape of TM in an image provides a geometric scale. In the case of S-type asteroid, the TM of low albedo and high emissivity (dark ball) should better be detected easily as a "hot sphere".

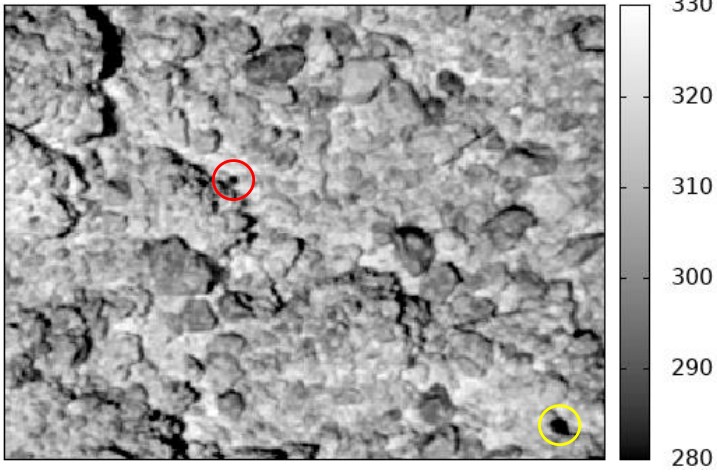

**Figure 6.** Close-up thermal image taken by TIR on 25 October 2018, during the descent operation of TD1-R3 Campaign. The width of this image is 6 m across. The Target Marker (TM, in the red circle) of the diameter of 10 cm was clearly detected in the thermal image as a "Cold Sphere" beside a 1 m-sized rugged boulder. This thermal image shows the surface is covered with several tens-of-centimeter- to meter-sized boulders larger than the TM and not covered with fine or small grained regolith materials. A very cold boulder (in yellow) is seen at the bottom-right in this thermal image. (Image: hyb2_tir_20181025_024712_l2.fit, ©JAXA).

### 5.4. Deployable Satellite or Rover Navigation

An impact experiment using the Small Carry-on Impactor (SCI) took place to expose the subsurface materials by forming a 10 m-sized artificial impact crater [29,30] on 5 April 2019 and collect the ejected materials on the asteroid surface that were sedimented around the artificial impact crater during the

second touchdown operation. Thermal images have successfully tracked the SCI moving from the spacecraft with the background against the asteroid surface after it was released at altitude of 500 m (see Figure 7). TIR detected a nutation of the SCI, and it was found to be sufficiently small to keep the impact point on Ryugu within a few tens of meters beside the target position on Ryugu. The optical camera was also used but it could not image the SCI and the asteroid surface at the same time without overexposure because of too higher reflectance of SCI compared to the low reflectance of the surface of the C-type asteroid. Thermal imaging has an advantage of simultaneous imaging of the SCI and the asteroid surface without any change of exposure time. This capability is useful and applicable to take an image of deployable small satellites in view of planetary surfaces, or to navigate the rovers moving on planetary surfaces.

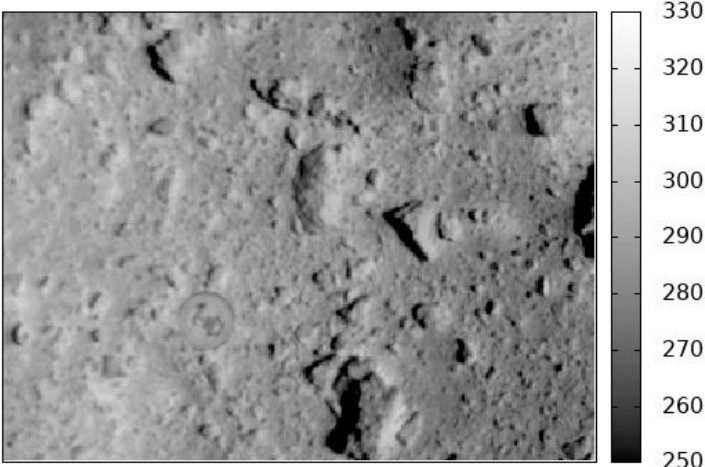

**Figure 7.** Thermal image of the Small Carry-on Impactor (SCI) released from Hayabusa2 spacecraft taken by TIR on 5 April 2019, at the altitude of 500 m. The shape and the attitude of the SCI was recognized in thermal images against the background of asteroid surface. A movement of SCI with its small nutation was observed in this image. The temperature scale is in Kelvin. (Image: hyb2_tir_20190405_015705_l2.fit, ©JAXA).

*5.5. Operation at Nighttime*

During the proximity phase, optical images observed only the sunlit surface of Ryugu, while thermal images observed almost the whole surface regardless of day and night. This capability is important and applicable to operate and work on the planetary surfaces in the night and in the shadowed area, if the surface temperature is higher than 160 K in the case of TIR on Hayabusa 2 [31]. For example, when a rover works on the polar regions of the Moon where there are many shadows of boulders, optical images need lights if it is used in the shadow, but suffers from overexposure from the sunlit surface within the image. A thermal imager can be used more easily even in the dark side of the Moon. Thermal images are also applicable for in situ resource use such as mining on near-Earth asteroids where day and night changes rapidly because of fast rotation.

*5.6. Site Selection and Safe Assessment*

TIR contributed to the Hayabusa2 mission purpose for providing information on the landing site selection for sampling and the release of landers, as well as the safe assessment during the descent operations [19]. TIR data is applicable to provide the global distribution of temperatures at any time and the highest temperature in a day for each site. After determining the thermal inertia derived from the diurnal temperature profiles, surface physical states are mapped to find the preferable sampling sites with existence of fine or small particles or the sites with base rock surfaces, or to avoid the hazardous sites with numerous boulders. In addition, the temperatures of local sites are predictable by comparison with numerical studies regarding the time of descent operations to assess safe landing.

*5.7. Guidance and Navigation Control of Spacecraft*

In the Hayabusa2 mission, the spacecraft is basically kept on the day side of Ryugu in the field of view of the wide angle camera (except for the escape from the ejecta excavated by the SCI impact), because the optical images should be used for its navigation control all the time. As an application to future missions, such as European Space Agency's Hera mission to asteroid Didymos [32], the spacecraft will be operated at the proximity of the asteroid, especially at the large solar phase angles (or SPE angles), where detailed navigation control is required even with the night-side area to precisely determine the center of the asteroid. Thermal image will be useful for the navigation and guidance control to include the night-side area. Precise positioning is needed to know the asteroid whole shape, which is easily detectable by TIR using the function of subtraction of dark frame images and binarization.

## 6. Concluding Remarks

During the asteroid proximity phase of Hayabusa2, thermography was applied for the first time in planetary missions. Thermal infrared imaging by TIR has been conducted through the mission to investigate C-type near-Earth asteroid 162173 Ryugu, for achieving both scientific and mission objectives [4,19,23] (see Supplementary Materials). Much progress has been made in asteroid science, offering a new insight for understanding planetary formation scenario and hypothesizing a highly porous nature of small bodies in the current and probably ancient Solar System [4]. Global, local, and close-up thermal images by the TIR have unveiled the thermophysical properties of the surface of Ryugu and provided us with new technical information and experiences. The observations of TIR on Hayabusa2 mainly from the stational home position for asteroid one-rotation are very helpful for data interpretation even for the boulder-rich rough surface, and are achievable for the mission success.

The limitation of this method using the TIR on Hayabusa2 or the instruments based on the same type of uncooled micro-bolometer array is the minimum detection temperature of 150 K or higher. This is mainly caused by the filtered wavelength range of 8–12 μm and the background thermal radiation from optics. The wavelength of typical commercial-based bolometer arrays ranges 8–14 μm, which is not substantially different from the current one. A two-dimensional detector to cover 20 μm or longer wavelengths would detect the thermal radiation from the colder planetary surface at a lower temperature such as the dark side of the Moon, any asteroid within the main belt, or the satellites of Jupiter. The thermal design to cool the optics might be feasible by passive radiative cooling or using a state-of-the-art active cooling system.

Thermography has been proven during the Hayabusa2 mission as a potentially useful and applicable tool for future space and planetary missions, for applications such as (1) optical navigation for approaching the distant target body, (2) detection of meteoroids or small satellites (ejected rocks) for hazardous protection of spacecraft, (3) target marker tracking for precise navigation and guidance for smart landing onto the surface of the target bodies, (4) tracking or navigating deployable small robots or landers against the backgrounds of planetary surfaces, (5) operations in the night or in the dark area, such as the dark side of the Moon, (6) landing site selection for determining the best site for sampling and for safety operations in hazardous protect or in thermal conditions, and (7) guidance and navigation of spacecraft around the asteroid or planetary satellites, such as orbiting the spacecraft. Some of these demonstrated technology should be applied in the future missions to the Moon, Mars, asteroids, comets, and other possible targets.

**Supplementary Materials:** TIR raw data and temperature converted data with ancillary data are available or will be available online at the Hayabusa2 data site: darts.isas.jaxa.jp/pub/hayabusa2/tir_bundle/browse/. The timing of archival for each data is basically one year after acquisition of the data.

**Funding:** This work is partially supported by the JSPS Kakenhi No. JP17H06459 (Aqua Planetology).

**Acknowledgments:** The author appreciate all the members of Hayabusa2 project and the people who contributed to its related development, operations, and scientific studies for fruitful discussions and technical supports. The author would give special thanks to the members of TIR teams.

**Conflicts of Interest:** The author declares no conflict of interest.

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
