# Peer review of "Thermography of Asteroid and Future Applications in Space Missions"

_applsci, doi:10.3390/app10062158_

Round 1

Reviewer 1 Report

The paper titled "Thermography of asteroid and future applications in space mission" by T. Okada describes the first use of thermography in space missions. In the body of the text, there is a review of the scientific results obtained by the TIR-S instrument on board of the Hayabusa 2 mission towards the 162173 Ryugu C-Type asteroid. Furthermore, a clear and well-described series of applications of TIR to successive and future space missions, it is also given.
The paper is well written and clear and this referee recommends its publication after that the following minor issue will be addressed:

- The paper lacks a brief description of the TIR - S instrument with its more important parameters, operational modes, download rate, etc., also if, as it is likely, the TIR instrument is fully described in a dedicated reference.
I think that a short paragraph could help the reader to be more aware of the results described in the paper.

Author Response

The paper titled "Thermography of asteroid and future applications in space mission" by T. Okada describes the first use of thermography in space missions. In the body of the text, there is a review of the scientific results obtained by the TIR-S instrument on board of the Hayabusa 2 mission towards the 162173 Ryugu C-Type asteroid. Furthermore, a clear and well-described series of applications of TIR to successive and future space missions, it is also given.

The paper is well written and clear and this referee recommends its publication after that the following minor issue will be addressed:

- The paper lacks a brief description of the TIR-S instrument with its more important parameters, operational modes, download rate, etc., also if, as it is likely, the TIR instrument is fully described in a dedicated reference.

--> The description of the TIR-S is shown at the beginning of the section 1 (Introduction), with the reference [1], which is the paper for detailed description of TIR. Here we added the table of TIR performance. And some other descriptions are added in the text for the readers to understand more easily.

- I think that a short paragraph could help the reader to be more aware of the results described in the paper.

--> The purpose of this paper is not focused on the science results achieved by TIR on Hayabusa2 (which is referred [4]), but on providing the potential applications to future space missions. However, the summary of results was added in Section 4.

Reviewer 2 Report

The topic of the manuscript is fascinating and shows the versatility of thermal infrared remote sensing methods. It gets clear, that the usage of this method is innovative and reasonable for different approaches on space missions. Especially the close-up thermal images, collected for the first time, show a huge benefit for innovative analysis of asteroid surfaces and materials in the future.

Suggestions for revisions:

The introduction starts with the detailed parameter of the thermal infrared imager (TIR), these should better be placed in chapter 3. There are only very little basics concerning infrared thermography and the application for rock and sediment properties as well as for concepts, such as albedo, emissivity, isolation, TIR view angles and tiu. Especially because of the descriptive character of the manuscript in some parts, the basics get more important for a comprehensible line of arguments. For a better understanding of your findings in TIR image analyses, please include these basics into your introduction and refer to them in your assumptions in a more analytical way. Also, the aim of this paper should get clearer by expressing open research questions or/and hypothesis at the end of the introduction.

If possible (if data are available and processed by now), include some data (of other instruments), numerical or statistical analyses and prove the assumptions in this way. An overview (table) of assumptions, generated by the thermal images (as the new method) in comparison to standard measurements could help assessing the infrared thermography at the approaches and applications here. You also could (additionally) refer to the results of other studies in more detail, such as Sugita et al. 2019, Mizuno et al. 2017, Okada et al. 2019…)

Please define the “temperature” as the infrared surface temperature at the beginning of the manuscript.

Please specify, that the relative infrared temperatures are important for your analysis. However, all of the infrared images need to have temperature scales (or a short explanation of the color scale), especially because the absolute temperature levels are only mentioned shortly in the text. Concerning to this, a figure, which shows the diurnal change of infrared temperatures (for 1 pixel or polygon of one surface location) as an overview would be reasonable, for example in chapter 4.3, where information about special day times are mentioned (noon in line 148).

The abbreviation “TIR” could probably irritate parts of the readership, because TIR also is the abbreviation for “thermal infrared” in general. Maybe there is an option for you to handle this “problem”.

The conclusions in chapter 5.2 refer to this special thermal infrared instrument, which was used. Please add the huge potential of other infrared sensors, for example with a higher resolution and/or include this in the conclusion as an outlook for the future.

A critical view on the thermal infrared method used here including method limitations or problems with image collection, design or evaluation should be included into this last chapter as well.

In detail:

Line 44, Figure 1: please choose a picture with a higher resolution Line 128, Figure 2: please include a temperature scale; a smaller temperature spectrum could distinguish the asteroids surface with a higher contrast; showing the sequence the other way round (in temporal order) would be more intuitive Line 132, Table 1: is a reasonable overview of the observation time scale. Please include explanations, why the shown parameter are important for your analysis and conclusions; showing the sequence the other way round (in temporal order) would be more intuitive Line 133: a short introduction to chapter 4 would be nice, not directly starting with 4.1 (applies also for other chapters) Line 142-144: Please explain, why this was your assumption before using the TIR data Line 146: explain the approach with “one-layer flat surface” and “subsurface materials” (line 262) more in detail, respectively define these terms Line 149: is the assumption, that Ryugu is consistent with very porous surface based on thermal insulation effects? If so please clarify this and include the relevant physical basics into the introduction Lines 155: the explanation concerning figure 3 should be placed in chapter 4.2 Line 174, Figure 3: please include a temperature scale; this figure could be extended with the geometrical facts of Ryugu (such as size, position of equator, et cetera) and the marks of crater and boulder position, which are mentioned in the text Line 183-184: please explain this assumption more in detail and concerning the thermal imaging method; which physical and methodical basics are used behind Line 206, Figure 4: please include a temperature scale Line 211, Figure 5: please include a temperature scale Line 245: please define the “smearing effect”; maybe another term should be used? Line 281, Figure 7: please include a temperature scale; an explanation for this figure in the text is missing. Please include, for example in line 263 Line 291: …of the moon where there are many shadows of… Line 322: the conclusions seems to be very short, concerning the great aspects and ideas of the work; please include more aspects or precise the experiences, mentioned here Line 377: Space Sci. Rev. reformat into italic type

Author Response

The topic of the manuscript is fascinating and shows the versatility of thermal infrared remote sensing methods. It gets clear, that the usage of this method is innovative and reasonable for different approaches on space missions. Especially the close-up thermal images, collected for the first time, show a huge benefit for innovative analysis of asteroid surfaces and materials in the future.

Suggestions for revisions:

The introduction starts with the detailed parameter of the thermal infrared imager (TIR), these should better be placed in chapter 3. There are only very little basics concerning infrared thermography and the application for rock and sediment properties as well as for concepts, such as albedo, emissivity, isolation, TIR view angles and tiu. Especially because of the descriptive character of the manuscript in some parts, the basics get more important for a comprehensible line of arguments. For a better understanding of your findings in TIR image analyses, please include these basics into your introduction and refer to them in your assumptions in a more analytical way. Also, the aim of this paper should get clearer by expressing open research questions or/and hypothesis at the end of the introduction.

>> This paper is prepared to start with a brief introduction of the TIR instrument, to show some facts/evidences of the observations of asteroid Ryugu and other deployed devices, and to provide the potential applications to future missions. The science results are shown in the other papers (e.g., ref. [4] and future papers). Information of instruments are already shown in the original version (e.g., ref. [1]) but some explanations and the table of TIR performance were added in the revised version.

If possible (if data are available and processed by now), include some data (of other instruments), numerical or statistical analyses and prove the assumptions in this way. An overview (table) of assumptions, generated by the thermal images (as the new method) in comparison to standard measurements could help assessing the infrared thermography at the approaches and applications here. You also could (additionally) refer to the results of other studies in more detail, such as Sugita et al. 2019, Mizuno et al. 2017, Okada et al. 2019…)

Please define the “temperature” as the infrared surface temperature at the beginning of the manuscript.

>> The comparison with numerical studies is conducted in the other papers (e.g., Ref. [4] and [26]) and the in-depth study with various parameters are now submitted to other journal, so that it is not shown in this paper. The science outputs of other instruments are referred (ref. [19-23, 27,28]), since this is focused on thermographic observations.

Please specify, that the relative infrared temperatures are important for your analysis. However, all of the infrared images need to have temperature scales (or a short explanation of the color scale), especially because the absolute temperature levels are only mentioned shortly in the text. Concerning to this, a figure, which shows the diurnal change of infrared temperatures (for 1 pixel or polygon of one surface location) as an overview would be reasonable, for example in chapter 4.3, where information about special day times are mentioned (noon in line 148).

>> That is a good suggestion, but the diurnal temperature profiles several sites are shown in the paper [4] and discussed in more detail in the paper [26] whose updated version is already submitted and in review in the other journal (by Shimaki et al.).

The abbreviation “TIR” could probably irritate parts of the readership, because TIR also is the abbreviation for “thermal infrared” in general. Maybe there is an option for you to handle this “problem”.

>> Yes, it may be confusing, but it usually happens that the often used abbreviations are adopted as the names of instruments, such as VIS, NIR, LIR, MAG, MGF, SEIS, MS, LIDAR, LRF, XRS, GRS, NS, so that the authors consider it no problem, if it is defined in the text.

The conclusions in chapter 5.2 refer to this special thermal infrared instrument, which was used. Please add the huge potential of other infrared sensors, for example with a higher resolution and/or include this in the conclusion as an outlook for the future.

>> This is out of scope of this paper, but the authors added the explanation why this type detector is used for Hayabusa2. That is due to the light-weighted, low power, and longevity during the long-time space cruise, which was added in the introduction. Specifications possibly useful in the future mission are briefly added in the Concluding Remarks

A critical view on the thermal infrared method used here including method limitations or problems with image collection, design or evaluation should be included into this last chapter as well.

>> We added the critical view in the last chapter, especially for the lower limit of temperature detection, so that it changed from Conclusions to Concluding Remarks.

In detail:

Line 44, Figure 1: please choose a picture with a higher resolution

>> As was also pointed out by the other reviewers, the author prepared the clearer photograph for Figure 1.

Line 128, Figure 2: please include a temperature scale; a smaller temperature spectrum could distinguish the asteroids surface with a higher contrast; showing the sequence the other way round (in temporal order) would be more intuitive.

>> The asteroid is smaller than a single pixel size in the earlier thermal images, so that the apparent temperature is observed as the lower value than the actual temperature of the asteroid surface. It is meaningless to show the temperature (or temperature scale) so that we do not show the temperature scale in this figure. The sequences of images were changed in temporal order according the suggestion by the reviewer-2.

Line 132, Table 1: is a reasonable overview of the observation time scale. Please include explanations, why the shown parameters are important for your analysis and conclusions; showing the sequence the other way round (in temporal order) would be more intuitive

>> The incidence and emergence thermal flux are inversely proportional to the square of the distance from the Sun and the asteroid, respectively. The solar phase angle is also important considering the surface roughness or irregular shape of asteroid, but the measurable angle is the Sun-Probe-Earth which is almost equal to the solar phase angle during the approach time. The description is briefly added in the text.

Line 133: a short introduction to chapter 4 would be nice, not directly starting with 4.1 (applies also for other chapters)

>> The authors added a brief comment for each of those chapters.

Line 142-144: Please explain, why this was your assumption before using the TIR data

>> The assumption should be valid on the Earth, under a 1-G gravity field condition, or on Mars or on the Moon, where rocks are nominally consolidated. It was a surprise that we found the unconsolidated boulders on Ryugu.

Line 146: explain the approach with “one-layer flat surface” and “subsurface materials” (line 262) more in detail, respectively define these terms

>> The authors added the explanation for these terms. The one-layer flat surface is the simplest model for thermal calculation. And the authors added the explanation on the subsurface materials which are those originally under the ground but excavated by the impacts of the liner of SCI to form a more than 10-meter scale crater, and consequently exposed from the spacecraft.

Line 149: is the assumption, that Ryugu is consistent with very porous surface based on thermal insulation effects? If so please clarify this and include the relevant physical basics into the introduction

>> We added the comments about the possibility of porous materials in the second paragraphs of the Introduction.

Lines 155: the explanation concerning figure 3 should be placed in chapter 4.2

>> Maybe this comment pointed out that Figure 3 should be placed in the same section where it is referred. The authors corrected it properly.

Line 174, Figure 3: please include a temperature scale; this figure could be extended with the geometrical facts of Ryugu (such as size, position of equator, et cetera) and the marks of crater and boulder position, which are mentioned in the text

>> The position of the largest crater Urashima and the largest boulder Otohime are indicated in the figure. Temperature scale is not shown because the precise calibration (for the 3rd version) is still on-going for interpreting the night area. This image is used to show even the night area is also detectable using TIR.

Line 183-184: please explain this assumption more in detail and concerning the thermal imaging method; which physical and methodical basics are used behind

>> The assumption is based on their temperatures which show almost the same as the large boulders, so that the surface materials must have the thickness of thermal skin depth, which corresponds to several centimeters.

Line 206, Figure 4: please include a temperature scale

>> The temperature scale is added, with some names of geologic features.

Line 211, Figure 5: please include a temperature scale

>> The temperature scale is added in Figures 5, 6, and 7. 

Line 245: please define the “smearing effect”; maybe another term should be used?

>> The author believes that “smearing and blooming effect” may be used, especially for optical camera using the CCD. This is the generic term for the optical detector so that no definition is needed but it is helpful for readers to add some explanation.

Line 281, Figure 7: please include a temperature scale; an explanation for this figure in the text is missing. Please include, for example in line 263

>> The authors added the link of Figure 7 in the text.

Line 291: …of the moon where there are many shadows of…

>> The author corrected the phrase according to the comment of Reviewer-2.

Line 322: the conclusions seems to be very short, concerning the great aspects and ideas of the work; please include more aspects or precise the experiences, mentioned here.

>> The authors added the descriptions of the critical view point of this method in the Concluding Remarks. The detection limit at 150K is the problem such as in case of the mission in the night side of the Moon.

Line 377: Space Sci. Rev. reformat into italic type

>> The author reformatted “Space Sci. Rev.” into italic type.

Reviewer 3 Report

Overall the work of the Hyabusa2 Thermal Infrared Imager described here is interesting and informative. I have no scientific concerns about the work, but the language is in need of revisions. Similarly, the image in Figure 1 is not particularly clear or legible -- it is difficult to make out the pieces described in the caption. A cartoon schematic would be very helpful in ascertaining what the authors are describing with more clarity.

Author Response

Overall the work of the Hyabusa2 Thermal Infrared Imager described here is interesting and informative. I have no scientific concerns about the work, but the language is in need of revisions. Similarly, the image in Figure 1 is not particularly clear or legible -- it is difficult to make out the pieces described in the caption. A cartoon schematic would be very helpful in ascertaining what the authors are describing with more clarity.

>> The clearer photograph of the TIR-S is prepared for Figure 1. The description of Figure 1 is updated to make the description clearer, as “The flight model of the sensor unit of the thermal infrared imager TIR on Hayabusa2 (TIR-S). Velcros are to attach the multi-layer insulators (the thermal blankets, not shown) which cover the sensor unit to stabilize the temperature during the operation. A baffled sunshade (not shown) is attached in front of the Germanium lens to avoid insolation. The aperture of the lens is 47 mm in diameter, and the stepping motor for the mechanical shutter is seen at the right hand of the lens. The scale is 10 cm (1 cm each)”.

Reviewer 4 Report

The paper “Thermography of Asteroid and Future Applications in Space Missions” is a good paper and worthy of publishing. I have minor comments that can help improve the quality of the paper.

General Comments:

A native English speaker should be invited to go over the document and correct small grammatical mistakes. This would greatly help the readability of the document. I invite the author’s to consider including Alan Harris in their reference list. His work is considerable on the work of thermal imaging of asteroids. For example: Harris, A. W., & Lagerros, J. S. V. (2002). Asteroids in the Thermal Infrared. In Asteroids III (Vol. 205, pp. 205–218). Retrieved from http://www.lpi.usra.edu/books/AsteroidsIII/pdf/3005.pdf

Specific Comments

L 44: needs higher resolution image. The scale in the image is not readable L 55: Initially, I took issue with writing “tiu” to represent the physical unit of thermal inertia. However, the authors are correct that they make heavy use of “tiu” and I find that it improves readability. I am fine with them keeping “tiu”. L 73: Throughout the document, check for consistency in writing Hayabusa 2 or Hayabusa2. This line reads Hayabusa2 where previous instances write Hayabusa 2. One version should be applied throughout the text. L 87: The word the authors wish to use is not “course” but “coarse”. Check other instances of usage as well (e.g. l 90) L 92: Using the acronym HP for “Home Position” is unnecessary and makes it harder for the reader. The authors should simply write “home position” in the three instances they use the expression in the text. L 211: Although I believe that I can identify the cold boulder the authors describe in the figure caption, it would be useful to be explicit and to mark the spot of interest in the figure. You could use a circle or an arrow to show which spot you mean. L 231: I have trouble understanding the meaning of this sentence. Consider rephrasing. L 231: Incorrect usage of the term “meteor”. A meteor is the phenomena of an asteroid burning up in the atmosphere. The authors should use the general term “asteroid” or, for smaller bodies, “meteoroid”. L 274: It is not entirely clear if the red circle in figure 6 marks the TM or the boulder. Ideally, the authors should mark both points of interest in the figure to be clear. L 318: Instead of “done” use “made” in this sentence.

Author Response

The paper “Thermography of Asteroid and Future Applications in Space Missions” is a good paper and worthy of publishing. I have minor comments that can help improve the quality of the paper.

General Comments:

A native English speaker should be invited to go over the document and correct small grammatical mistakes. This would greatly help the readability of the document. I invite the author’s to consider including Alan Harris in their reference list. His work is considerable on the work of thermal imaging of asteroids. For example: Harris, A. W., & Lagerros, J. S. V. (2002). Asteroids in the Thermal Infrared. In Asteroids III (Vol. 205, pp. 205–218). Retrieved from http://www.lpi.usra.edu/books/AsteroidsIII/pdf/3005.pdf

>> The author added the description of the importance of asteroid science and related physical properties to be observed by the thermal imaging. The paper by Harris and Lagerros in Asteroid III is a kind of textbook of asteroid thermal infrared observations and thermal modeling, so that the author added the paper as a reference.

Specific Comments

L 44: needs higher resolution image. The scale in the image is not readable.

>> The same comment from the Reviewer 3. The clearer photograph is used in Figure 1 and the caption is updated, with the description of the scale.

L 55: Initially, I took issue with writing “tiu” to represent the physical unit of thermal inertia. However, the authors are correct that they make heavy use of “tiu” and I find that it improves readability. I am fine with them keeping “tiu”.

>> The author appreciated the reviewer-4 for the understanding of the use of “tiu” as the unit of thermal inertia. The basic problem is non-existence of the unit of thermal inertia.

L 73: Throughout the document, check for consistency in writing Hayabusa 2 or Hayabusa2. This line reads Hayabusa2 where previous instances write Hayabusa 2. One version should be applied throughout the text.

>> Hayabusa2 is the formal name of the mission, so that we have to use “Hayabusa2”. The description only in L73 was mistakenly used, so that it was corrected.

L 87: The word the authors wish to use is not “course” but “coarse”. Check other instances of usage as well (e.g., L 90)

>> The author corrected the word in the text at L87 and L90, thanks to the reviewer-4.

L 92: Using the acronym HP for “Home Position” is unnecessary and makes it harder for the reader. The authors should simply write “home position” in the three instances they use the expression in the text.

>> The author agreed with the idea not to use the “HP” or “Home Position”, but to use the “home position”.

L 211: Although I believe that I can identify the cold boulder the authors describe in the figure caption, it would be useful to be explicit and to mark the spot of interest in the figure. You could use a circle or an arrow to show which spot you mean.

>> The author agreed with the comment by the Reviewer-4 to explicitly show the cold boulder in the thermal image with a red circle.

L 231: I have trouble understanding the meaning of this sentence. Consider rephrasing.

>> The author rewrote the sentence as “As was mentioned in 5.1, the fact that an object is seen as the only bright spot in thermal image enables to detect the meteoroids or the orbiting moons which encounter nearby the spacecraft.”

L 231: Incorrect usage of the term “meteor”. A meteor is the phenomena of an asteroid burning up in the atmosphere. The authors should use the general term “asteroid” or, for smaller bodies, “meteoroid”.

>> The author agreed with the comment of Reviewer-4 that the term “meteor” should be changed to “meteoroid”.

L 274: It is not entirely clear if the red circle in figure 6 marks the TM or the boulder. Ideally, the authors should mark both points of interest in the figure to be clear.

>> The object in the red circle is the TM. The boulder is shown in yellow in Figure 6.

L 318: Instead of “done” use “made” in this sentence.

>> The author agreed with the comment by the Reviewer-4 comment and “done” was changed to “made”.

Round 2

Reviewer 2 Report

Thanks for implementing most of the comments, made in review 1. You improved the quality of the paper, additionally by adapting other reviewers’ comments. The rejection of some comments are well founded and science-base. I now recommend the paper to be published in the new version of the manuscript in the Applied Science journal.